# Inhibitory Effects of Nisin and Gallium (III) Nitrate Hydrate on Planktonic and Adhered Cells and Implications for the Viable but Non-Culturable State

**DOI:** 10.3390/microorganisms13020276

**Published:** 2025-01-26

**Authors:** Valeria Poscente, Luciana Di Gregorio, Roberta Bernini, Annamaria Bevivino

**Affiliations:** 1Department for Sustainability, Sustainable Agri-Food Systems Division, ENEA, Italian National Agency for New Technologies, Energy and Sustainable Economic Development, Casaccia Research Center, 00123 Rome, Italy; valeria.poscente@enea.it (V.P.); luciana.digregorio@enea.it (L.D.G.); 2Department of Agriculture and Forest Sciences, University of Tuscia, 01100 Viterbo, Italy; roberta.bernini@unitus.it

**Keywords:** biofilms, antimicrobial treatments, nisin, gallium (III) nitrate hydrate, viable but non-culturable (VBNC) state, flow cytometry

## Abstract

Effective antimicrobial and biofilm control strategies require an understanding of the differential effects of antimicrobial agents on the viability and culturability of microbial cells. A viable but non-culturable (VBNC) state, a survival strategy of non-spore-forming bacteria in response to adverse conditions, poses a significant challenge for public health and food safety. In the present study, we investigated the antimicrobial and antibiofilm effects of nisin and gallium (III) nitrate hydrate against the Gram-positive strain *Lactiplantibacillus plantarum* subsp. *plantarum* DSM 20174 and the Gram-negative strain *Pseudomonas fluorescens* ATCC 13525, respectively. Both strains were chosen as model systems for their relevance to food and clinical settings. Culture-based methods and flow cytometry (FCM) were used to evaluate the culturability and viability of both planktonic and sessile cells, providing insights into their physiological response to antimicrobial treatment-induced stress at different concentrations (100, 250, 350, and 500 ppm). The findings highlight the strain-specific action of nisin on *L. plantarum* and the promising antibiofilm effects of Ga (III) against *P. fluorescens*. This study underscores the promising potential of FCM as a powerful tool for high-throughput analyses of antimicrobial efficacy, providing valuable insights into developing targeted biofilm control strategies for food safety and clinical applications.

## 1. Introduction

Strategies to reduce biofilm formation are increasing, with natural antimicrobial agents emerging as promising alternatives [1]. Biofilms exhibit enhanced resistance to antimicrobial agents and host immune responses compared to planktonic cells, contributing to chronic infections, persistent industrial surface contamination, and diminished antimicrobial treatment efficacy [2,3]. During biofilm formation, bacteria can enter a “viable but non-culturable” (VBNC) state, where metabolic activity persists but culturability is lost [4]. This state confers increased resistance to physical, chemical, and antibiotic treatments, posing a great threat to food safety and public health, especially for foodborne pathogens that retain virulence after the “resuscitation” process [5,6]. Many of the antimicrobial agents normally used for the eradication of microorganisms in food processing plants, hospitals, and domestic environment are not very effective on VBNC bacteria [7,8]. Most importantly, as entering a VBNC state is a physiological adaptation to stress, the exposure to antimicrobial treatment can also induce the VBNC state [9]. Detecting VBNC cells necessitates advanced microbiological techniques such as flow cytometry (FCM), fluorescence microscopy, fluorescence in situ hybridization (FISH), real-time polymerase chain reaction (qPCR), and advanced imaging methods, crucial for overcoming the limitations of the traditional culture-based methods that are unable to detect subtle physiological changes. The use of FCM allows for more accurate assessments of bacterial viability, metabolic activity, and response to antimicrobial treatments [10,11,12,13,14,15,16]. Natural antimicrobial peptides like nisin, a bacteriocin produced by *Lactococcus lactis*, have shown potential in food safety applications. Nisin was approved by the Food and Drug Administration (FDA) in 1988 and is “generally recognised as safe” (GRAS) for use in foods due to its broad-spectrum antimicrobial properties. It has been demonstrated to inhibit biofilm formation through mechanisms such as the disruption of cell membranes and interference with quorum sensing [17,18]. Nisin demonstrated antimicrobial efficacy against several bacteria, including *Staphylococcus aureus*, *Listeria monocytogenes*, *Alicyclobacillus acidoterrestris*, *Clostridium* and *Bacillus* spores, and even inhibits some Gram-negative strains [19,20,21,22]. Nevertheless, repeated exposure to increasing nisin concentrations has been shown to induce resistance in bacteria such as *Lacticaseibacillus casei*, *Streptococcus thermophilus*, *L. monocytogenes*, and *S. aureus*. Even small populations of resistant bacteria in long-shelf-life foods can proliferate, leading to foodborne outbreaks and spoilage [22,23]. However, while increasing attention is being focused on exploring the effects of nisin on different strains, its impact on model systems such as *Lactiplantibacillus plantarum* remains limited [24,25,26,27,28,29,30,31]. It has been shown that the class IIb bacteriocin plantaricin A in the presence of *Fructilactobacillus sanfranciscensis* DPMMA174 and *L. plantarum* DPPMA20 actually facilitated the formation of biofilms by *L. plantarum* DC400 [31,32]. Understanding nisin’s impact on this strain could provide insights into its dual role in promoting beneficial biofilms and preventing pathogenic biofilms in food production systems [31,33,34].

In the last decade, gallium-based compounds have emerged as potential antimicrobial agents [35]. Indeed, gallium (III) nitrate hydrate [Ga (III)] ions act as iron mimics within bacterial cells, disrupting crucial iron-dependent processes like DNA replication and respiration [36,37]. This interference inhibits bacterial growth and virulence, making it challenging for bacteria to develop resistance to Ga (III) without compromising iron uptake [37]. The biological activity of Ga (III) ions has received increasing attention because of its current uses in clinical practices, mainly in trials focusing on lung function improvement in cystic fibrosis patients with chronic *Pseudomonas aeruginosa* lung infections, as approved by FDA [36,38,39,40]. Consequently, gallium-based compounds are emerging as potential next-generation antibiotics due to their efficacy against resistant bacteria interfering with quorum sensing pathways and reducing extracellular polymeric substance (EPS) synthesis [41,42,43,44]. Compared with organic antibacterial agents, inorganic antibacterial agents exhibit stronger environmental stability, and they generally show strong antimicrobial activity, even in small amounts [45]. Furthermore, the low toxicity and broad-spectrum antibacterial properties of Ga (III) make it a promising candidate for ensuring food safety without contributing to bacterial resistance [45]. In this study, a *Pseudomonas fluorescens* model strain [46], was chosen as a common food spoilage bacterium to explore the antibiofilm effects of Ga (III), considering its relevance to biofilm-associated spoilage in the food industry. In the present work, we aimed to investigate the effect of nisin and Ga (III) on the culturability and viability of planktonic and sessile cells of *L. plantarum* subsp. *plantarum* DSM 20174, and *P. fluorescens* ATCC 13525, respectively. Culture-based and flow cytometry approaches were used to investigate the culturability and viability after the exposure of each microorganism to various concentrations of the antimicrobial compounds (100, 250, 350, and 500 ppm). A comprehensive PubMed search, using keywords such as *Gallium nitrate*, *Pseudomonas fluorescens*, VBNC, bacterial viability, bacterial persistence, biofilms, adherent cells, and antimicrobial activity, revealed no studies investigating the impact of Gallium (III) nitrate hydrate on the bacterial vitality of *Pseudomonas fluorescens*, particularly in VBNC states. Our study addresses this gap through innovative approaches such as flow cytometry. The lack of studies highlights the novelty of our research, which explores the antimicrobial effects of nisin and Ga (III) on VBNC and biofilm-associated cells through advanced methodologies. These findings establish a foundation for advanced approaches to managing bacterial physiological responses, offering significant potential for antimicrobial packaging systems, infection control, and surface pretreatment applications.

## 2. Materials and Methods

### 2.1. Strains and Cultural Conditions

In this study, the Gram-positive *L. plantarum* subsp. *plantarum* DSM 20174 and the Gram-negative *P. fluorescens* ATCC 13525 reference strains were stocked at −80 °C in 15% (*v*/*v*) glycerol until use. From frozen stocks, *L. plantarum* and *P. fluorescens* were revitalised overnight in 5 mL of sterile de Man, Rogosa and Sharpe broth (MRS) (Merck KGaA, Darmstadt, Germany), and tryptic soy broth (TSB) (Merck KGaA, Darmstadt, Germany) for the Gram-positive and Gram-negative strains, respectively. *L. plantarum* was grown under static conditions at 30 °C, while *P. fluorescens* was grown in shaking conditions (180 rpm) at 28 °C. From the revitalised cultures, the strains were sub-cultured first in 10 mL and then in 30 mL of sterile medium. A starting inoculum concentration of 6 Log CFU/mL was applied to conduct the tests [34].

### 2.2. Nisin and Gallium (III) Nitrate Hydrate Stock Preparation

Nisin from *Lactococcus lactis* (ref. N5764) and gallium (III) nitrate hydrate (ref. 289892) were purchased from Sigma-Aldrich (St. Louis, MO, USA) and stored at 4 °C until use. A 10% (*w*/*v*) stock solution of each compound was prepared in sterile distilled water and filtered through a 0.2 µm pore size polycarbonate membrane. To test the antimicrobial activity, test tubes were prepared by inoculating an equal ratio (1:1) of antimicrobial solution and bacterial culture (6 Log CFU/mL) in a final volume of 5 mL of MRS broth and TSB for *L. plantarum* and *P. fluorescens*, respectively. This setup achieved the final antimicrobial concentrations of 100, 250, 350, and 500 ppm. Untreated controls were included by using sterile distilled water (0 ppm) for each tested condition.

### 2.3. In Vitro Activity of Nisin and Gallium (III) Nitrate Hydrate Against Planktonic and Sessile Cells in the 96-Well Microtiter Plates

To evaluate the inhibitory effect of nisin and gallium (III) nitrate hydrate on planktonic and adhered cells of *L. plantarum* and *P. fluorescens,* respectively, experiments were carried out using 96-well plates (Falcon™ 96-well flat-bottom microplates). A volume of 0.2 mL of bacterial cultures in each condition, with and without the antimicrobial compounds at different concentrations (0, 100, 250, 350, and 500 ppm), was transferred from test tubes into separate 96-well plates and incubated for 24 h before the planktonic and sessile fraction analysis. The microtiter plates containing *L. plantarum* strain were incubated statically at 30 °C, whereas those containing the *P. fluorescens* strain were maintained at 28 °C [34]. Inoculum-free controls were present in each plate.

#### 2.3.1. Quantitative Analyses of Biofilm by Using Crystal Violet Staining

The Crystal Violet (CV) assay was used in this study to assess cell attachment and viability by staining attached cells, which were subsequently detached from the culture plates, as described in the study by Bragonzi et al. [47] and Poscente et al. [34]. Briefly, after 24h of growth, the planktonic fractions were transferred to new microtiter plates for the enumeration of planktonic cells (see Section 2.3.2), while the attached cells were rinsed three times with 0.2 mL of phosphate-buffered saline (PBS) to remove non-adherent and weakly adherent bacteria. After air-drying the plates for 30 min, 0.2 mL of 1% (*w*/*v*) CV was added to each well, followed by a 20 min incubation at room temperature. The excess CV was removed by washing the wells three times with 0.2 mL of PBS. The dye bound to the sessile cells was dissolved by adding 0.2 mL of ethanol solubilization solution (95% *v*/*v*) and transferred to a new 96-well plate for the absorbance measurement at 595 nm using a Promega™ GloMax^®^ automated reader. The degree of CV staining directly correlates with the amount of cell biomass attached to the plate. To account for background absorbance, OD readings from sterile medium, dye, and ethanol were averaged and subtracted from all test values. Experiments were performed in triplicate and repeated in three independent experiments.

#### 2.3.2. Culturability Assay of Planktonic and Sessile Cells

To enumerate the planktonic fraction, the unadhered cells suspensions, previously transferred to new microtiter plates, were quantified by plating 0.1 mL of 10-fold serial dilutions on MRS (*L. plantarum*) and TSA (*P. fluorescens*) agar plates, as previously described [34,47]. The colony forming units (CFU) per mL were evaluated after 24 h of incubation at 30 °C and 28 °C for *L. plantarum* and *P. fluorescens*, respectively. Planktonic cell growth was also investigated through the turbidity assay by the optical density measurement at 595 nm.

The enumeration of the related sessile (adhered) cell fraction was performed as described in the study by Bragonzi et al. [47] and Poscente et al. [34]. In brief, each well was rinsed three times with 0.2 mL of phosphate-buffered saline (PBS, pH 7.4) to remove non-adherent and weakly adherent bacteria. Then, the biofilm was recovered by scraping the surface of each well with 1 mL of PBS and vortexed for 30 s. The adhered cells thus recovered were quantified by plating serial dilutions of biofilm samples on MRS and TSA media, as described above for the planktonic cells. To ensure the complete detachment of the bacteria, a CV (1%) assay was performed on each of the wells scraped, and absorbance was determined at 595 nm. All data were obtained from three independent experiments performed in triplicate [34,47].

### 2.4. Viability Assay of Planktonic and Sessile Cells by FCM Analysis

The CytoFLEX S flow analyzer (Beckman Coulter, Flow Cytometry, Milan, Italy) was used in this study for the viability investigation of planktonic and sessile fraction, as described previously [48]. The same samples used for culturability assay (both planktonic and sessile cells) were also analysed by FCM analysis. Briefly, from the 96-well plate, each cell suspension was subjected to a 100-fold dilution in PBS and a double-staining procedure with SYTO24 (5 µM) and Propidium Iodide (15 µM) was performed for the membrane integrity evaluation. Samples were analysed by using the blue laser (ext. 488 nm) and the bandpass filters collecting fluorescence emissions at BP525/40 and BP675/30 for the SYTO24 and PI fluorescence, respectively. Viable cells (SYTO positive) exhibited a green signal, while damaged/dead cells (PI positive) showed red fluorescence, corresponding to the degree of damage. The SYTO24 vs. PI dot plot allowed the different cell subpopulations (viable, damaged, and dead cells) to be distinguished according to a different ratio of green/red fluorescence reflecting the state of cell viability. Data were processed as percentages referring to the total number of events recorded, excluding the background signal. The acquisition of 50.000 events per sample was carried out with a slow flow rate (10 µL/min) and by applying FSC 10.000 threshold settings. Regions of interest were defined in relation to the untreated control (live cells, positive controls) and autoclaved bacterial culture (dead cells, negative controls). The parameters were acquired with a logarithmic scale and analysed using CytExpert software v. 2.3 (Beckman Coulter Flow Cytometry, Milan, Italy). Microspheres of 2.5 µm diameter (Alignflow™ for Blue Lasers, Thermo Fisher Scientific Life Science Solutions, Milan, Italy) were used as internal reference standards. Results were obtained from three replicates from three independent experiments, and the standard deviation was always <5%.

### 2.5. Statistical Analysis

©GraphPad Prism Software (version 10.0.3) was used for the statistical analysis. The data are presented as the mean ± standard deviations (SD) based on triplicates from three independent experiments. Data were compared using two-way ANOVA with Dunnett’s multiple comparison tests and Tukey’s pairwise test at *p* < 0.05, which was considered statistically significant (95% confidence interval).

## 3. Results and Discussion

### 3.1. Nisin Effect Against L. plantarum DSM 20174

Nisin antimicrobial action against the planktonic fraction of the *L. plantarum* strain showed promising results even at the lowest concentration of 100 ppm (Figure 1A). These findings align with the known antibacterial properties of nisin, a lantibiotic produced by *L. lactis*, which is effective against a wide range of Gram-positive bacteria, including controlling biofilm formation by foodborne pathogens (e.g., staphylococci, bacilli, and clostridia) and antibiotic-resistant bacteria like methicillin-resistant *S. aureus* (MRSA) [49,50]. Due to its interaction with anionic lipids on the bacterial cell membrane, the antimicrobial action of nisin results in membrane disruption by forming pores that lead to the efflux of ions, dissipation of the proton-motive force, ATP hydrolysis, and therefore, loss of viability and cell lysis [51]. Moreover, there is also a steadily growing number of engineered nisin peptides that exhibit enhanced functionalities (activity and/or stability) which make them more attractive from a clinical perspective [50,52]. Results from the plate count analysis and turbidity assay consistently showed significant growth reductions and turbidity inhibition in all tested conditions compared to the control (*p* < 0.05). Treatments displayed similar reduction values, with the greatest logarithmic reduction observed at 350 ppm (Figure 1A). TFCM analysis was conducted to investigate cell physiology as a promising tool able to rapidly provide information on bacterial susceptibility and antimicrobial resistance, facilitating the development of more effective therapeutic and preventive strategies for food and clinical applications [10,48,53]. The results aligned with the plate count findings, showing 17.8% viable cells at 100 ppm. Viability decreased at higher concentrations, with increased background signals indicating cell membrane disruption by the antimicrobial agent (Figure 1A and Figure 2). It is important to highlight that the damaged cellular fraction was not detectable in significant values (<6%) at all tested concentrations, demonstrating that nisin has a bactericidal effect rather than a bacteriostatic effect against *L. plantarum*, even at low concentrations. The sessile fraction showed greater resistance to treatments with a lower logarithmic reduction than its planktonic counterpart in all tested conditions. Indeed, biofilms are intricate structures composed of polysaccharides, proteins, and extracellular DNA, each playing key roles in adhesion, protection, and nutrient exchange. These components facilitate the formation of microcolonies that are more resistant than planktonic cells, presenting challenges in both food and clinical environments [51,54,55]. Despite this, significant logarithmic reductions were noticed in all treatments compared to the control (*p* < 0.05) (Figure 1A). Turbidity data were consistent with plate count ones (Figure 1A,B). FCM analysis revealed no substantial differences in cellular physiology across the different tested concentrations (Figure 2). Internal cellular distribution did not change significantly compared to the control; while 58% of cells were alive, 13% were damaged cells, and 29% were dead cells. A distinct cell subpopulation within the sessile fraction was observed and identified in the dead cell gate by positive and negative controls. This likely occurred due to the interaction of the antimicrobial agent with the EPS matrix [56] (Figure 2). The results suggest the potential application of nisin in innovative antimicrobial packaging solutions to enhance the food safety and shelf-life of fresh products, as well as in clinical settings to prevent biofilm formation on medical devices, thereby reducing infection risks [34,57,58,59].

### 3.2. Gallium (III) Nitrate Hydrate Effect Against P. fluorescens ATCC 13525

Plate count results for *P. fluorescens* planktonic cells subjected to Ga (III) treatment demonstrated significant antimicrobial effects starting at a 250 ppm concentration (*p* < 0.05), with greater logarithmic reductions observed at 350 and 500 ppm (Figure 3A). Data obtained from the turbidimetric analysis were consistent with those from plate counts; even at the 500 ppm treatment, a reduced turbidity was observed (Figure 3B). Flow cytometry analysis provided insights into cell physiology as follows: despite similar logarithmic reductions for samples treated with 350 ppm and 500 ppm (3.34 and 3.39, respectively), at 350 ppm, most cells were viable (66%), whereas at 500 ppm, a greater bactericidal effect was observed (18% damaged cells, 40% dead cells) (Figure 4). A greater effect on the *P. fluorescens* sessile fraction compared to its planktonic counterpart was found (Figure 1A and Figure 3A). Ga (III) disrupts iron metabolism, which is crucial for biofilm formation and maintenance, affecting both planktonic and sessile bacteria by interfering with essential cellular functions and inducing dormancy states, thereby overcoming the protective function of the EPS matrix [41]. Its activity is promising for clinical applications where pre-treating medical devices with Ga (III) coatings could prevent biofilm formation and reduce infection rates. Similarly, in the food industry, Ga (III) coatings could inhibit microbial growth on surfaces that come into contact with food, ensuring product safety [41]. Plate count data related to the sessile fraction showed significant logarithmic reductions for all tested concentrations, and especially at 500 ppm, a total loss of cultivability was observed (Figure 3A,B). On the other hand, the flow cytometry dot plots showed how the physiology of the cell population changed in response to the tested concentrations (Figure 4). Starting from the lowest gallium (III) concentration of 100 ppm, where an internal cellular distribution of 22% viable cells, 16% damaged cells, and 62% dead cells was observed, the treatment at 250 ppm proved more effective, bringing the fraction of dead cells to 76% (Figure 4). Treatment at 350 ppm showed an increase in background signal because of the high degree of cellular impairment (Figure 4). A substantial difference in culturability and viability was observed after the 500 ppm treatment, where FCM revealed 38% viable cells showing that the treatment induced the cells into the VBNC state (Figure 4). Bacterial cells commonly respond to environmental stressors, like antimicrobial compounds, by losing their ability to form colonies in standard culture media and entering into a VBNC state, although cells can retain the ability to conduct fundamental metabolic processes, including respiration, nutrient assimilation, and gene expression [5,60,61,62]. Due to its multi-targeting antimicrobial activity against drug-resistant strains, Ga (III) has garnered significant attention as a novel antibacterial strategy. The results regarding Ga (III), particularly in relation to the food environment, are innovative and interesting. However, they need to be confirmed using Gram-positive models or pathogenic species [35]. The induction of a VBNC state by a high concentration of antimicrobial compounds [9] could pose a serious threat to human health, as the play an active role in relapsing infections and antibiotic failure. In the clinical laboratory, VBNC cell survival and resuscitation is much harder to identify as VBNC cells are not detectable using standard culture-based methods. Currently, several methodologies, such as molecular techniques and metabolism-focused and staining techniques, can be used to determine bacterial viability in the VBNC state [60,63]. While molecular techniques are based on individual and global gene expression detection in non-culturable cells, metabolism-based techniques search for metabolites or enzyme activities, generally using methodologies with colorimetric results, biosensors, or arrays. Alternatively, staining techniques are mainly based on the detection of electron transport system activity and cytoplasmic membrane integrity [60]. In this context, flow cytometry proves to be a rapid technique with a high-throughput potential for the investigation of the cell physiological state using only fluorescent probes with different permeability to the cell membrane depending on the state of cell damage [10,48].

## 4. Conclusions

This study underscores the effectiveness of nisin and gallium (III) nitrate hydrate as promising solutions for addressing biofilm-associated challenges in both food safety and clinical contexts. As model strains for Gram-negative and Gram-positive microorganims, *Lactiplantibacillus plantarum* subsp. *plantarum* DSM 20174 and *Pseudomonas fluorescens* ATCC 13525 were used to evaluate the antimicrobial and antibiofilm effect of nisin, a bacteriocin, and gallium (III) nitrate hydrate, respectively. *L. plantarum* was selected as model strain due to its well-known relevance in biofilm formation and antimicrobial resistance. Recent studies have also demonstrated its usefulness as a monolayer system for evaluating the combined efficacy with antimicrobials against foodborne pathogens and spoilage microorganisms [34]. Conversely, *P. fluorescens* was chosen as the Gram-negative model strain given its prominent role in biofilm-related spoilage issue within the food industry. 

Notably, nisin demonstrated a significant impact on the culturability and viability of *L. plantarum* planktonic cells, while Ga (III) proved more effective against adhered cells, suggesting its potential application in innovative antimicrobial packaging systems to enhance food safety and in clinical environments to prevent biofilm formation on medical devices, thereby reducing the risk of infections. The use of flow cytometry allowed us to reveal the effects of the antimicrobial treatments on bacterial physiology and their role in inducing the VBNC state. This highlights the critical need for advancing flow cytometry techniques to ensure the accurate assessments of antimicrobial efficacy. Planktonic cells, VBNC cells, and the biofilm cells of bacteria can coexist in food or food processing, posing serious challenges to public health and safety. Future research should prioritise elucidating the influence of nisin and gallium (III) nitrate hydrate on quorum sensing modulation and the biofilm formation process. Further insights into these mechanisms could pave the way for more effective strategies to prevent biofilm-associated infections, including the pre-treatment of surfaces with gallium (III) nitrate hydrate coatings in both the food industry and medical settings. Additional studies are needed to investigate the synergistic action of nisin and gallium (III) nitrate hydrate on biofilm-VBNC bacteria and the planktonic VBNC of Gram-negative bacteria, as well to extend the study of gallium (III) nitrate hydrate on opportunistic human pathogens [64,65]. Before the application of nisin and gallium (III) nitrate hydrate in food or clinical sectors, to ensure the safe application of both compounds in their respective fields, a comprehensive biosafety assessment is needed to evaluate the potential cytotoxic and genotoxic effects of nisin and Ga(III) on mammalian cells at the concentrations recommended in the present study.

In conclusion, this study poses a valuable foundation for the development of advanced strategies aimed at preventing and controlling biofilm formation, ultimately contributing to improved food safety and patient care outcomes.

## Figures and Tables

**Figure 1 microorganisms-13-00276-f001:**
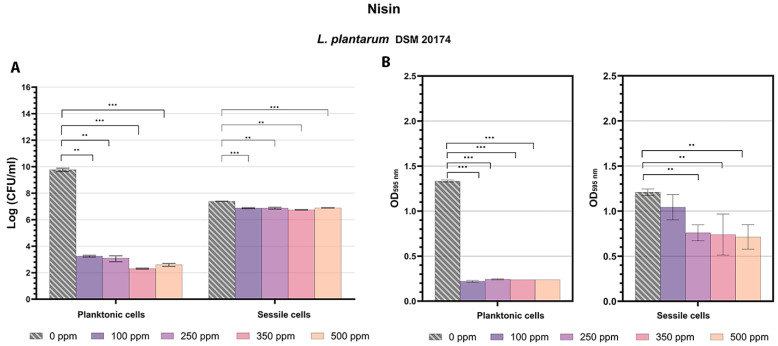
Effect of nisin (100 ppm, 250 ppm, 350 ppm, and 500 ppm) against planktonic and sessile cells of *L. plantarum* DSM 20174 after 24 h of incubation. (**A**) Histograms indicate the Log CFU/mL values obtained by standard plate count methods after 24 h of incubation compared to control (0 ppm). (**B**) Histograms indicate the turbidimetric analysis results (OD 595 nm) of planktonic cells (left side) and crystal violet results (OD 595 nm) of sessile cells (right side) after antimicrobial treatments compared to control (0 ppm). All data are presented as the mean ± standard deviations (SD) based on triplicates from three independent experiments. (**) *p* < 0.01, (***) *p* < 0.001.

**Figure 2 microorganisms-13-00276-f002:**
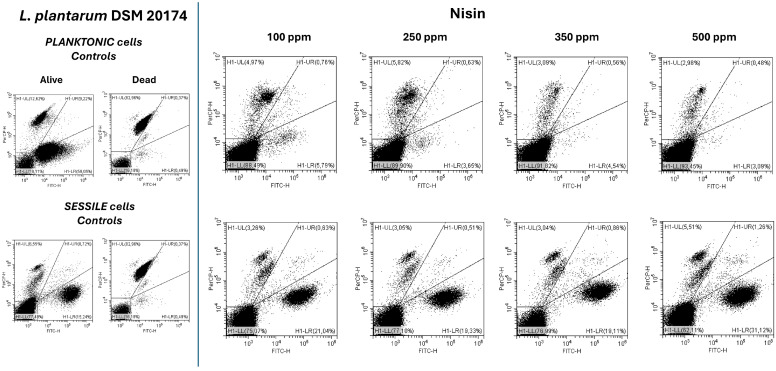
FCM analysis of *L. plantarum* DSM 20174 planktonic and sessile cells treated with nisin (100 ppm, 250 ppm, 350 ppm, and 500 ppm) after 24 h of incubation grown. Results are represented by a double-staining dot plot (SYTO24/PI) related to the planktonic (on top) and sessile fraction (below) compared to the untreated controls. H1-LL, unstained debris; H1-LR, intact cells/viable cells (SYTO24-positive); H1-UR, injured cell population; H1-UL, permeabilized/dead cells (PI-positive).

**Figure 3 microorganisms-13-00276-f003:**
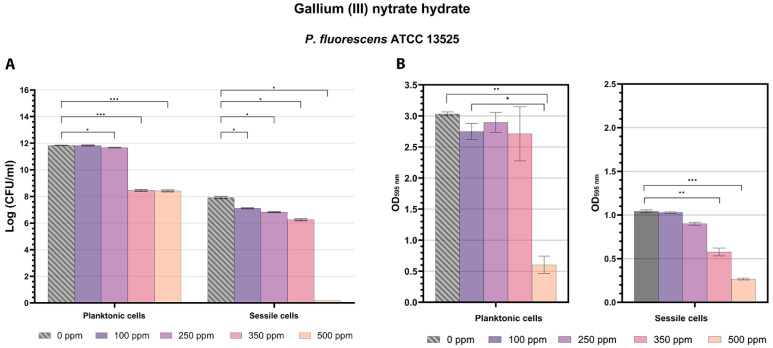
Effect of gallium (III) nitrate hydrate (100 ppm, 250 ppm, 350 ppm, and 500 ppm) against planktonic and sessile cells of *P. fluorescens* ATCC 13525 after 24 h of incubation. (**A**) Histograms indicate the Log CFU/mL values obtained by standard plate count methods after 24 h of incubation compared to control (0 ppm). (**B**) Histograms indicate the turbidimetric analysis results (OD 595 nm) of planktonic cells (left side) and crystal violet results (OD 595 nm) of sessile cells (right side) after antimicrobial treatments compared to control (0 ppm). All data are presented as the mean ± standard deviations (SD) based on triplicates from three independent experiments. (*) *p* < 0.05, (**) *p* < 0.01, (***) *p* < 0.001.

**Figure 4 microorganisms-13-00276-f004:**
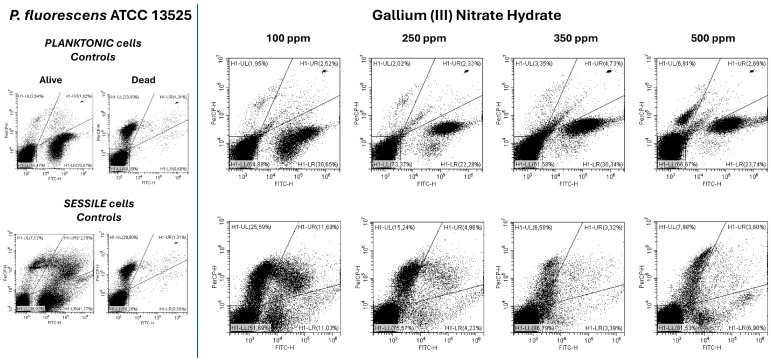
FCM analysis of *P. fluorescens* ATCC 13525 planktonic and sessile cells treated with gallium (III) nitrate hydrate (100 ppm, 250 ppm, 350 ppm, and 500 ppm) after 24 h of incubation. Results are represented by a double-staining dot plot (SYTO24/PI) related to the planktonic (on top) and sessile fraction (below) compared to the untreated controls. H1-LL, unstained debris; H1-LR, intact cells/viable cells (SYTO24-positive); H1-UR, injured cell population; H1-UL, permeabilized/dead cells (PI-positive).

## Data Availability

The original contributions presented in this study are included in the article. Further inquiries can be directed to the corresponding author.

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
