# Peer review of "Inhibitory Effects of Nisin and Gallium (III) Nitrate Hydrate on Planktonic and Adhered Cells and Implications for the Viable but Non-Culturable State"

_microorganisms, 2025, doi:10.3390/microorganisms13020276_

Round 1
Reviewer 1 Report
Comments and Suggestions for Authors
The manuscript is a well-written study that provides valuable insights into the effects of nisin and gallium nitrate on planktonic and adhered cells of two model bacterial strains.
However, the study would benefit from investigating the long-term effects of exposure, the potential for resistance development, and the potential of cells to recover from VBNC state after exposure to the tested compounds.
Author Response
REVIEWER 1 – MAIN COMMENT
The manuscript is a well-written study that provides valuable insights into the effects of nisin and gallium nitrate on planktonic and adhered cells of two model bacterial strains. However, the study would benefit from investigating the long-term effects of exposure, the potential for resistance development, and the potential of cells to recover from VBNC state after exposure to the tested compounds.
Authors’ response: We are grateful to the Reviewer for his/her valuable comment and suggestion. Investigating the long-term effects of exposure and the ability of cells to recover from the VBNC state after exposure to the tested compounds are all critical aspects that we have carefully considered for future studies. Specifically, we aim to address the potentially hazardous implications of the resuscitation process by employing advanced methodologies, such as flow cytometry and flow cell sorting. These approaches will allow us to monitor and characterize cellular recovery dynamics with high precision as well as will permit us to evaluate the potential for resistance development. We believe this line of research will substantially enhance the understanding and impact of our findings.
Reviewer 2 Report
Comments and Suggestions for Authors
The reviewed article is an original experimental study of the antimicrobial and antibiofilm effects of nisin, a bacteriocin, on the Gram-positive strain Lactiplantibacillus plantarum subsp. plantarum DSM 20174, and gallium (III) nitrate hydrate, a novel antimicrobial agent, on the Gram-negative strain Pseudomonas fluorescens ATCC 13525. The title of the article reflects the content, the aim is clearly stated, and the conclusions summarize the results well. I believe that the reviewed work will be improved by:
I. Clarification in the Materials and Methods section:
1. How was the sterility of the solutions ensured? It is not enough to use only sterile water to prepare the solutions; the starting compounds in this case must also be sterile.
2. From the description (Lines 144-146) it is not clear how planktonic cells were selected; it is stated that the wells were washed with a buffer solution to remove non-adherent and weakly adherent cells.
3. The biofilm removal method needs clarification (Line 147).
4. Lines 152-153 - Was the crystal violet method used in this case? If not, is it acceptable and scientifically reliable to compare the results of determining the optical density without the use of the dye (for planktonic cells) and with its use (for sessile cells)?
5. Describe in the Materials and Methods section what was used as a positive and negative control (Lines 221-222).
II. Correction of some technical errors, in particular the presence of some abbreviations that are not used further in the text, etc. (see the article file).
III. Clarification of Line 350 - Data Availability Statement: Not applicable. – Is this really so? After all, this is experimental work with new scientific data.
After making corrections, the article can be published.

Author Response
REPLY TO REVIEWER 2
REVIEWER 2 – MAIN COMMENT
The reviewed article is an original experimental study of the antimicrobial and antibiofilm effects of nisin, a bacteriocin, on the Gram-positive strain Lactiplantibacillus plantarum subsp. plantarum DSM 20174, and gallium (III) nitrate hydrate, a novel antimicrobial agent, on the Gram-negative strain Pseudomonas fluorescens ATCC 13525. The title of the article reflects the content, the aim is clearly stated, and the conclusions summarize the results well. I believe that the reviewed work will be improved by:
I. Clarification in the Materials and Methods section:
- How was the sterility of the solutions ensured? It is not enough to use only sterile water to prepare the solutions; the starting compounds in this case must also be sterile.
Authors’ response: Thanks for the comment. To address this concern, we have added specific details on how the sterility of the solutions was ensured in the revised manuscript (lines 163–164), improving the overall understanding of paragraph 2.2 (lines 160-169). We believe this addition clarifies the procedures we followed to ensure sterility throughout the preparation and use of the solutions.
- From the description (Lines 144-146) it is not clear how planktonic cells were selected; it is stated that the wells were washed with a buffer solution to remove non-adherent and weakly adherent cells.
Authors’ response: Thanks for the comment. As suggested, we have clarified the procedure in the revised manuscript (see sub-paragraph 2.3.1 lines 186–187 and 2.3.2 lines 316-322).
- The biofilm removal method needs clarification (Line 147).
Authors’ response: Thanks for the comment. Accordingly, we revised the sub-paragraph 2.3.2 improving the biofilm removal method (lines 323-321). We followed the procedure previously published in Bragonzi et al. 2012 and Poscente et al. 2023.
Bragonzi, A.; Farulla, I.; Paroni, M.; Twomey, K.B.; Pirone, L.; Lorè, N.I.; Bianconi, I.; Dalmastri, C.; Ryan, R.P.; Bevivino, A. Modelling co-infection of the cystic fibrosis lung by Pseudomonas aeruginosa and Burkholderia cenocepacia Reveals influences on biofilm formation and host response. PLoS One 2012, 7(12): e52330. doi:10.1371/JOURNAL.PONE.0052330.
Poscente, V.; Gregorio, L. Di; Costanzo, M.; Nobili, C.; Bernini, R.; Garavaglia, L.; Bevivino, A. Lactiplantibacillus plantarum monolayer enhanced bactericidal action of carvacrol : biofilm inhibition of viable foodborne pathogens and spoilage microorganisms. Front. Microbiol. 2023, 14:1296608. doi: 10.3389/fmicb.2023.1296608
- Lines 152-153 - Was the crystal violet method used in this case? If not, is it acceptable and scientifically reliable to compare the results of determining the optical density without the use of the dye (for planktonic cells) and with its use (for sessile cells)?
Authors’ response: Thanks for the comment. We improved the sub-paragraphs 2.3.1 and 2.3.2, removing the sentence “Planktonic cells growth was also investigated through the turbidity assay by the optical density measurement at 595 nm” (former lines 152-153) and added it in the lines 320-322. In our study, we measured the optical density of planktonic cells (without the use of the dye), in contrast for sessile cells, we employed the crystal violet staining method to assess biofilm formation. In addition, to determine the amount of each bacterium attached to the microtiter plates, we performed viable counts of bacteria detached from the wells of polystyrene plates (sessile cells) after an overnight incubation. Although these methods differ in terms of their approach to measurement, the results always considered the appropriate controls for both conditions. Turbidimetric analysis (for planktonic cells) and crystal violet staining (for sessile cells) were conducted in parallel with plate counts. This allowed us to establish a direct correlation between the observed growth and the signals obtained from each method. As a result, we ensured a reliable comparison of the growth and viability of planktonic and sessile cells, making the results scientifically valid.
- Describe in the Materials and Methods section what was used as a positive and negative control (Lines 221-222).
Authors’ response: Thanks for the comment. Regions of interest were defined in relation to the positive (untreated controls, live cells) and negative controls (autoclaved bacterial culture) using a double-staining procedure [SYBR Green I (using a 1:10,000 dilution of the stock reagent) and Propidium Iodide (PI) 10 μg/mL]. The viable cells exhibited a green signal (SYBR positive) while the damaged/dead cells showed an orange- red fluorescence (PI positive). All the controls used in the 96-well microtiter plates were indicated in the Materials and Methods section (lines 168-169, 180, 359-350).
II. Correction of some technical errors, in particular the presence of some abbreviations that are not used further in the text, etc. (see the article file).
Authors’ response: Thanks for the comment. We have revised the entire manuscript and made the suggested corrections, removing technical errors and abbreviations that were not further used in the text. We believe these changes have improved the clarity and readability of the manuscript.
III. Clarification of Line 350 - Data Availability Statement: Not applicable. – Is this really so? After all, this is experimental work with new scientific data.
Authors’ response: Thanks for the comment. We agree that it is important to ensure accessibility to experimental data. Accordingly, we revised the statement (lines 691-692) to clarify that the data generated during the study are included in the article, and further inquiries can be directed to the corresponding author.
After making corrections, the article can be published.
Authors’ response: We thank the Reviewer for his/her valuable comments and constructive remarks that improved the manuscript. We have taken the comments on board to improve and clarify the manuscript.

Reviewer 3 Report
Comments and Suggestions for Authors
The article "Inhibitory effect of nisin and gallium (III) nitrate hydrate on planktonic and adhesive cells and the effect on the state of VBNC" is devoted to a relevant topic. Results obtained by the authors are scientifically novel and have high applied potential. Text of the manuscript is well structured, written consistently and logically. Introduction is excellently written; it masterfully leads the reader to the main problem of the article. Methods are described in sufficient detail. Authors provided a rationale for the scope of the experiment, the conclusions made are fully consistent with the results obtained during the experiment.
My only comment will be the absence in the text of the manuscript of materials on measuring the biosafety (at least cyto- and genotoxicity upon contact with mammalian cells) of nisin and gallium (III) nitrate hydrate in the concentrations that authors recommend using in practice. Without any experiments or analysis of the results of other authors on the biosafety of nisin and gallium (III) nitrate hydrate lead to their use in the food industry and clinical practice prematurely.
Author Response
REPLY TO REVIEWER 3
REVIEWER 3 – MAIN COMMENT
The article "Inhibitory effect of nisin and gallium (III) nitrate hydrate on planktonic and adhesive cells and the effect on the state of VBNC" is devoted to a relevant topic. Results obtained by the authors are scientifically novel and have high applied potential. Text of the manuscript is well structured, written consistently and logically. Introduction is excellently written; it masterfully leads the reader to the main problem of the article. Methods are described in sufficient detail. Authors provided a rationale for the scope of the experiment, the conclusions made are fully consistent with the results obtained during the experiment.
My only comment will be the absence in the text of the manuscript of materials on measuring the biosafety (at least cyto- and genotoxicity upon contact with mammalian cells) of nisin and gallium (III) nitrate hydrate in the concentrations that authors recommend using in practice. Without any experiments or analysis of the results of other authors on the biosafety of nisin and gallium (III) nitrate hydrate lead to their use in the food industry and clinical practice prematurely.
Authors’ response: We thank the Reviewer for his/her positive comment. We agree that a comprehensive assessment of the biosafety of nisin and gallium (III) nitrate hydrate is crucial prior to their application in the food industry or clinical practice. It is well known that nisin is considered safe by the World Health Organization (WHO) and the Food and Drug Administration (FDA-United States), being used initially as a food additive (de Arauz et al. Nisin biotechnological production and application: A review. Trends Food Sci. Technol. 2009;20:146–154. doi: 10.1016/j.tifs.2009.01.056). In previous studies, no cytotoxicity was observed (see Dos Santos et al. Bacterial Cellulose Membranes as Carriers for Nisin: Incorporation, Antimicrobial Activity, Cytotoxicity and Morphology. Polymers (Basel). 2022 Aug 26;14(17):3497. doi: 10.3390/polym14173497; Gao et al. Production of nisin-containing bacterial cellulose nanomaterials with antimicrobial properties through co-culturing Enterobacter sp. FY-07 and Lactococcus lactis N8. Carbohydr. Polym. 2021;251:117131. doi: 10.1016/j.carbpol.2020.117131). On the other hand, in a recent work (Zainodini et al. Asian Pac J Cancer Prev. 2018 Aug 24;19(8):2217-2222. doi: 10.22034/APJCP.2018.19.8.2217), nisin was found to induce cytotoxicity and apoptosis in human asterocytoma cell line, opening a new window for establishment promising approaches with the concept of anti-cancer therapy by nisin in the future. As suggested by the Reviewer, for its use in the food industry by incorporation nisin into a film or in the clinical sector, the cyto- and genotoxicity of nisin at the concentrations recommended in the present study need to be verified. Regarding gallium (III) nitrate hydrate, we acknowledge that its biosafety profile requires further investigation for potential cyto- and genotoxicity particularly at the concentrations recommended in our study. Its administration as food preservatives/additives or in the clinical sector can have a significant effect on human health. In a preliminary Phase 1 trial, a single five-day infusion with gallium nitrate, improved the lung function of CF patients with a chronic P. aeruginosa infection, showing no signs of toxicity (Goss et al. Gallium disrupts bacterial iron metabolism and has therapeutic effects in mice and humans with lung infections. Sci Transl Med. 2018 Sep 26;10(460):eaat7520. doi: 10.1126/scitranslmed.aat7520). Indications of safety and efficacy of intra-venous gallium nitrate (GaN) administration against Pseudomonas aeruginosapneumonia as well as the Ga(III) encapsulation into hyaluronic acid/chitosan nanoparticles have been found (Costabile et al. Boosting lung accumulation of gallium with inhalable nano-embedded microparticles for the treatment of bacterial pneumonia. International Journal of Pharmaceutics,Volume 629,2022,122400, https://doi.org/10.1016/j.ijpharm.2022.1224009, and cited references).
In the revised manuscript, we improved the Discussion section (see lines 671-676) highlighting the need for further research to assess and/or exclude the potential cytotoxic and genotoxic effects of both nisin and gallium (III) nitrate hydrate, at the recommended concentrations, contributing to ensuring their safe and effective applications.